# Control of Gelation and Properties of Reversible Diels–Alder Networks: Design of a Self-Healing Network

**DOI:** 10.3390/polym11060930

**Published:** 2019-05-28

**Authors:** Beata Strachota, Adama Morand, Jiří Dybal, Libor Matějka

**Affiliations:** 1Institute of Macromolecular Chemistry, Academy of Sciences of the Czech Republic, Heyrovsky Sq. 2, 162 06 Prague 6, Czech Republic; beata@imc.cas.cz (B.S.); dybal@imc.cas.cz (J.D.); 2Sigma Clermont, Campus des Cezeaux, 63178 Aubiere, France; Adama.Morand@sigma-clermont.fr

**Keywords:** reversible covalent network, Diels–Alder reaction, network formation, self-healing of networks

## Abstract

Reversible Diels–Alder (DA) type networks were prepared from furan and maleimide monomers of different structure and functionality. The factors controlling the dynamic network formation and their properties were discussed. Evolution of structure during both dynamic nonequilibrium and isothermal equilibrium network formation/breaking was followed by monitoring the modulus and conversion of the monomer. The gelation, postgel growth, and properties of the thermoreversible networks from tetrafunctional furan (F4) and different bismaleimides (M2) were controlled by the structure of the maleimide monomer. The substitution of maleimides with alkyl (hexamethylene bismaleimide), aromatic (diphenyl bismaleimide), and polyether substituents affects differently the kinetics and thermodynamics of the thermoreversible DA reaction, and thereby the formation of dynamic networks. The gel-point temperature was tuned in the range *T_gel_* = 97–122 °C in the networks of the same functionality (F4-M2) with different maleimide structure. Theory of branching processes was used to predict the structure development during formation of the dynamic networks and the reasonable agreement with the experiment was achieved. The experimentally inaccessible information on the sol fraction in the reversible network was received by applying the theory. Based on the acquired results, the proper structure of a self-healing network was designed.

## 1. Introduction

The polymer networks form basis of materials with a broad range of applications. In comparison to thermoplastics, they show a better environmental resistance, mechanical, thermal, and generally high performance properties for application in structural materials, coatings, adhesives or in composite systems. However, contrary to thermoplastics, they are not reprocessable and recyclable. Nowadays, the crucial requirement for new polymer materials involves a lengthening of a material service time and possibility of recycling, while keeping their excellent properties. Self-healing of polymer network materials is a great challenge for both basic and applied research.

The covalent reversible networks meet these requirements. They are suitable for different applications, as reshapeable thermosets, for bonding/de-bonding adhesives, for self-healing, etc. The corresponding polymer material shows an intermedium behavior between the classical irreversible networks, thermosets and thermoplastics [1,2]. Unlike the permanent covalent networks, the dynamic networks display at low experimental frequencies the transition to the flow region of a viscoelastic liquid. This is a manifestation of the transient character of the dynamic networks, whose state is dependent on the time scale [1,3]. The formation of reversible networks is based on the application of reagents undergoing the reversible reactions, such as disulfide exchange [4], reaction of isocyanate and imidazole [5], ester exchange [6], etc. The most common is the thermoreversible Diels–Alder (DA) cycloaddition of a diene with a dienophile, mainly the reactions of furan and maleimide compounds [7,8,9,10,11] (Scheme 1).

The DA reaction to form the cyclohexene adduct and the reverse retro-DA (rDA) reaction results in the temperature-dependent thermodynamic equilibrium. At high temperatures, the rDA reaction dominates and the equilibrium is shifted towards the monomers. Formation of the reversible covalent networks from DA type monomers or cross-linking of polymers containing pendant functional groups is widely studied [1,12,13,14,15,16,17,18,19,20]. Reversibility of cross-linking is usually investigated by sol–gel analysis [21] or by DMA and rheology experiments [3,12,13,22,23,24]. The formation of the thermoreversible networks is characterized by the critical temperature for gelation, *T_gel_*. Due to an incomplete equilibrium conversion, these networks contain a sol fraction, which is an important attribute of a network structure. This parameter, however, is experimentally difficult to be determined in the reversible networks. So far, no corresponding investigation was performed according to our knowledge, despite the fact that this factor could limit the materials application.

The self-healing of thermoreversible networks is based on the dynamic breaking and reforming of cross-links. Due to thermally induced reversibility of cross-linking/decross-linking, the local mobility in the network is temporarily increased in the network by heating above some threshold temperature. This makes it possible welding of a mechanical damage such as crack/scratch by accelerated diffusion of chains. Restoration of the initial network structure and properties is then attained by cooling, promoting the reformation of the DA adduct/cross-link.

In this paper, we describe the formation of thermoreversible DA networks from furan and maleimide monomers of functionality *f* = 2–6. The goal of the work consists in the finding the factors enabling control of gelation and properties of the dynamic networks. The detailed study is focused mainly on the networks of bismaleimides with the tetrafuran monomer prepared by modification of Jeffamine D2000 with furan groups. The Jeffamine-based tetrafunctional monomer with a variable spacer length between furan functionalities was previously used by Scheltjens and Diaz [3,15]. In our case, the networks are formed by the reaction of the tetrafuran with bismaleimides of different structure, involving aliphatic, aromatic and polyether type substituents. The effect of a monomer structure, as well as functionality and composition of the monomer mixture or homogeneity of the system on gelation, network build-up, and its dynamics are investigated.

The kinetics of the reversible DA reaction and its thermodynamics, as well as the temperature dependence of the equilibrium conversion, were followed by FTIR. Both dynamic and isothermal structure evolution during network formation, including gelation, was determined rheologically by monitoring shear modulus. The equilibrium gelation temperature *T_gel_* was evaluated and the main parameters governing the network formation were discussed. In addition, the theory of branching processes was used for description of gelation and development of structure.

The structure of bismaleimides affects kinetics and thermodynamics of the DA reaction. As a result, the gelation, stability, and cross-linking density and dynamics of the thermoreversible networks were shown to be controlled by structure of maleimide monomers, in addition to functionality and composition of monomers. The gelation temperature can be tuned in a broad range. The experimental results are in a good agreement with the theoretical prediction. It makes possible a deeper insight into the mechanism of network formation. By using the theory of network formation we get the information on the sol fraction in the reversible networks. The theory is thus a valuable tool to provide the information on this crucial aspect of dynamic networks.

Based on these results, the design of a self-healing reversible network with the optimum structure was proposed.

## 2. Materials and Methods

### 2.1. Materials 

Furfurylamine (FA), furfuryl glycidyl ether (FGE), hexamethylenediamine, tris(2-aminoetheyl)amine, and the Jeffamines (polyoxypropylene)diamine D2000 (*M* = 1960 g/mol) and (polyoxypropylene)triamine T3000 were received from Sigma-Aldrich, Prague, Czech Republic. The bismaleimide monomer 1,1′-(methylenedi-4,1-phenylene) bismaleimide (DPBMI) was obtained from Sigma-Aldrich, Prague, Czech Republic and poly(oxypropylene)bismaleimides PPO3BMI (*M* = 408 g/mol) and PPO30BMI (*M* = 2350 g/mol) were received from Specific Polymers, Castries, France. The other monomers were synthesized. 

### 2.2. Synthesis of Monomers

#### 2.2.1. Tetrafunctional Furan Monomer Based on D2000 Jeffamine-F4D2000 

The Jeffamine D-2000 was functionalized according to [25] by furan through the epoxy-amine reaction with furfuryl glycidylether (FGE) at 90°C for 2 days (see Scheme 2). Hydroquinone (1% of the mixture weight) was added as inhibitor of polymerization of furfuryl groups. The structure of the product was confirmed by FTIR-ATR (attenuated total reflectance) and ^1^H-NMR spectroscopy (see Appendix A). The conversion was 95% and the corresponding weight average functionality thus was *f_w_* = 3.84.

The hexafunctional monomer F6T3000 and the trifunctional monomer F3FAFGE were prepared in the same way by the reaction of FGE with T3000 Jeffamine and furfurylamine (FA), respectively. Analysis is given in Appendix A.

#### 2.2.2. N,N’-hexamethylenebismaleimide (HBMI) 

HBMI was prepared in three steps according to Lacerda et al. [26] (i) synthesis of furan–maleic anhydride DA adduct (FMA) (3,6-epoxy-1,2,3,6-tetrahydrophthalic anhydride) [27] (Scheme 3), (ii) reaction of FMA with hexamethylene diamine, and (iii) splitting of the product by the rDA reaction. Analysis is given in Appendix A. 

(i) Maleic anhydride was solubilized in anhydrous diethyl ether and furan was added. The solution was kept under stirring at room temperature until the formation of crystals, indicating the adduct formation. The crystals were isolated by filtration, washed with diethyl ether to remove any unreacted maleic anhydride and dried under reduced pressure. Yield: ca. 80 %. ^1^H NMR and FTIR characterization is given in Appendix A.

(ii) Solution of hexamethylenediamine in methanol was added to a solution of FMA in methanol with drops of trimethylamine (Scheme 4). The solution was heated to reflux at 55–60 °C for 3 days. After cooling of the mixture a precipitate was formed. The suspension was filtered and the solid residue washed with cold methanol.

(iii) The intermediate product was dissolved in toluene and the retro-Diels–Alder reaction was carried out 8 h at 120 °C, setting the final product (Scheme 4).

#### 2.2.3. Tris(2-maleimidoethyl)amine (TMIEA)

The trifunctional maleimide monomer (TMIEA) was synthesized in the two-step process [28] (Scheme 5). The reaction of tris(2-aminoethyl)amine with FMA, in stoichiometric conditions, gave the corresponding trifunctional DA adduct (T-FMA), followed by deprotection using retro-DA reaction to release the furan groups. 

(i) A solution of FMA in methanol was cooled to 0 °C and then solution of amine in methanol was added dropwise over a period of 30 min. The mixture was heated to reflux for 6 h and then it was concentrated and left to crystallize at 0 °C. The obtained pale-yellow crystals were filtered and washed with methanol. The operation was done several times with the filtrate. The solid product was dried under reduced pressure at 35–40 °C (yield: 11%). Characterization is given in Appendix A.

(ii) The T-FMA intermediate was dissolved in toluene and submitted to the deprotective retro-DA reaction, over 7 h of reflux (120–130 °C). After solvent evaporation the residue yellowish solid was dissolved in ethyl acetate and passed over silica column. Pure solid was obtained after removal of the ethyl acetate (yield: 80%). Characterization is given in Appendix A.

The applied monomers are given in Scheme 6.

### 2.3. Preparation of Networks

The furan and maleimide monomers were reacted mostly in stoichiometric ratios for 24 h at 70 °C. The crystalline maleimides DPBMI and HBMI were dissolved in dichloromethane to obtain a homogeneous mixture prior the DA reaction. The dichloromethane was slowly removed during the reaction at 25 °C for 24 h and then under vacuum at 70 °C. The initially homogeneous F4D2000–PPO3BMI and F4D2000–PPO30BMI mixtures were reacted without a solvent. The inhibitor hydroquinone is present in the mixture in order to avoid homopolymerization of maleimides.

### 2.4. Methods

#### 2.4.1. FTIR Measurements 

ATR FTIR spectra were collected on a Thermo Nicolet Nexus 870 spectrometer (Thermo Fisher Scientific Inc., Waltham, Massachusetts, USA) purged with dry air and equipped with a mercury–cadmium–telluride (MCT) detector cooled with liquid nitrogen. The spectra were acquired on a horizontal micro-ATR Golden Gate unit (SPECAC) equipped with a diamond prism and a controlled heated top plate. All the ATR FTIR spectra were processed by the advanced ATR correction using the OMNIC^TM^ software.

The chemical reaction between maleimide and furan is manifested by changes of band intensities in several regions of the FTIR spectra. However, the bands are mostly overlapped and it is difficult to unambiguously distinguish contributions from individual chemical constituents. The kinetics of the reaction was followed by the intensity changes of the band at 695 cm^−1^, which is assigned to the skeletal vibration (breathing mode) of the ring in maleimide and it disappears after fully completing the reaction. For the determination of the band parameters the deconvolution of the spectrum in the region 780–660 cm^−1^ was performed. The acquired spectra were fitted with the Voigt function using the Peak Resolve routine of the OMNIC^TM^ software, which also allowed evaluating the integral intensity of the bands. All the parameters were allowed to vary during the fittings. 

#### 2.4.2. Dynamic Mechanical Analysis (DMA) and Dynamic Rheometry

DMA was performed with a rheometer ARES G2 apparatus (TA Instruments, New Castle, Delaware, USA). The temperature dependence of the complex shear modulus of rectangular samples (30 × 8 × 1 mm) was measured at a heating rate of 3 °C/min by using oscillatory shear deformation at a frequency of 1 Hz. The shear deformation was kept in the range 0.01 to 3% during the temperature sweep in order to keep a proper torque.

The rheology experiments were performed with ARES in the parallel plate geometry (6 mm in diameter, TA Instruments, New Castle, Delaware, USA). Three types of measurements were performed. (i) Temperature sweeps at heating and cooling rates of 0.5–5 °C/min, at a frequency of shear deformation of 1 Hz; (ii) isothermal time sweeps at a frequency of deformation of 1 Hz,; and (iii) frequency sweeps in the range of 0.002–200 rad/s. The strain was varying in the range of 50 to 2% in order to keep a sufficient torque while to stay in the linear viscoelastic region and to prevent breakage of the formed structure. 

Determination of the point of gelation by chemorheology is based on the application of the power law describing the rheological behavior of the system in the critical state [29], *G‘*(*ω*) *~ G“*(*ω*)*~ ω^n^*. Accordingly, the loss factor tan δ is independent of the experimental frequency at the gel-point. The multifrequency sweep (range of frequencies: 1–64 Hz) is used to follow the dependence of tan δ on frequency during the reaction. Appendix A shows the evolution of tan δ curves at different frequencies during the reaction. The time of gelation corresponds to the frequency independent loss factor. Due to a more clear presentation of the structure evolution during the reaction, we have evaluated the gel point by a simplified way from the crossover of *G‘* and *G*“(i.e., at tan δ = 1) at a frequency of 1 Hz. It was proved that in the critical state at the gel-point, our systems display the frequency independent critical loss factor value, (tan δ)_gel_ = 1.5 (see Appendix A). The difference in determination of the gel-point by both procedures, multifrequency sweep or critical (tan δ)_gel_ = 1.5 and the crossover of G’ and G”, is negligible, being almost in the range of an experimental error. The application of this simple procedure of the gel-point determination is required also from the fact that the multifrequency sweep cannot be used in case of the fast reaction (at high temperatures) due to a change of dynamic quantities during the sweep.

## 3. Results and Discussion

### 3.1. Kinetics and Thermodynamics of Diels–Alder Reversible Reaction

We studied kinetics of the DA reaction of tetrafunctional furan monomer F4D2000 with three maleimide monomers. The aromatic DPBMI, aliphatic HBMI, and the bismaleimide PPO3BMI with poly(oxypropylene) short chain were used and the effect of a maleimide structure was determined. It is known [30,31] that addition of the appropriate substituents in furan and maleimide compounds can significantly affect reactivity and reversibility of the cycloaddition reaction. The kinetics of the reversible reaction at different temperatures was followed by FTIR both in the solution (6% in dioxane) and in the bulk system. The maleimide conversion was monitored only because the conversions of maleimide and furan are equal during the reaction at the stoichiometric composition and a limited temperature. The temperature dependence of the forward DA and the reverse retro-DA (rDA) reaction rates, as well as the equilibrium position were evaluated. The data are given in the Appendix A. The reaction was followed in the temperature range of 20 to 120 °C in order to avoid the homopolymerization of maleimides occurring at a higher temperature [15,32]. 

The reaction rate of cycloaddition to form the DA cyclohexene adduct increases with increasing temperature, however the equilibrium conversion decreases due to the rDA reaction dominating at a high temperature (see Figure 1). In the solution, the equilibrium conversion at room temperature reaches the high value, *α_e_* = 0.82–0.90, while at 120 °C the reversible reaction is shifted to the monomers and *α_e_* = 0.47–0.50. The determined activation energies, *E_a_,_DA_* = 32–38 kJ/mol, *E_a_,_rDA_* = 65–81 kJ/mol, are slightly lower compared to literature data [3,12,15,33], however they are in accordance with the data obtained using polymeric systems [19,20]. Moreover, the DA reaction is known to involve the effect of stereochemistry. Both exo- and endo-isomer adducts are formed differing in the rate of formation and thermodynamic stability [33,34]. This effect, however, was not taken into account in the evaluation of the kinetics because the FTIR measurement does not allow to distinguish between these isomers.

The reaction kinetics in bulk was followed in a solvent-free mixture of monomers in case of the homogeneous system F4D2000–PPO3BMI. The maleimides HBMI and DPBMI are crystalline and the initial monomer mixtures are heterogeneous. In order to eliminate the effect of phase separation, the procedure starting from a cross-linked network was used. 

In the case of HBMI the cured network is homogeneous due to the reaction blending and the mixture with DPBMI is homogenized by melting of the bismaleimide at 120 °C. The cured networks were heated for 30 min at 120 °C to promote the rDA reaction and to break down the DA covalent bonds. The partially split up structure was cooled down to a particular temperature and the kinetics of the isothermal bonds reformation was followed by FTIR as illustrated in Figure 2. The reaction starts from a relatively high conversion, due to the incompletely broken structure at 120 °C. The reaction in bulk was found to show higher rate constants than in a dioxane solution (See Appendix A). Taking into account the rate constants and the concentration of functional groups in bulk systems, the reaction rates of maleimides in the network decrease in the series for the forward DA reaction are HBMI > DPBMI > PPO3BMI, and for the reverse rDA reaction are DPBMI > PPO3BMI > HBMI, respectively.

The equilibrium conversion in the solid state in the networks and its temperature dependence was determined by annealing the corresponding network under isothermal conditions at different temperatures. The results reveal that the equilibrium position is dependent on the maleimide structure. Moreover, the conversions are higher in solid networks compared to the reaction in solution (see Figure 3), as found also by other groups [3,15,33]. The DA cycloadduct is relatively stable at temperatures below 70 °C. The retro-DA reaction is insignificant at *T* < 70 °C, mainly in solid networks. Particularly the HBMI involving network exhibits the high equilibrium conversion. In this case, the cyclohexene adduct is stable up to very high temperatures. The equilibrium conversion *α_e_* > 0.90 at *T* = 70 °C and it is 0.60 even at 120 °C. 

The results are in a general agreement with the work of Boutelle et al. [30] describing the effect of maleimide substituents on the furan–maleimide DA reaction. Both the N-alkyl and N-phenyl substituents increase exergonicity of the reaction compared to the unsubstituted maleimide. The alkyl substitution shows the greatest effect as the more negative reaction free energy was reported compared to the N-phenyl substituents and nonsubstituted maleimides. Moreover, the electrondonating group, the methoxy (ether group) on the furan, increases the stability of the adduct with respect to methyl (alkyl group), while this substitution on electron poor maleimide exhibits an opposite effect. These substitutions, however are also predicted to decrease the transition state barriers, not hindering thus the reversibility of the reaction in spite of favoring the DA adduct formation. In accordance with these calculations, the applied maleimides show high conversions decreasing in the series (see Figure 3); *α_e_*(HBMI) > *α_e_*(DPBMI) ~ *α_e_*(PPO3BMI).

The DA cross-linking of the partially broken down structure shows an important difference from the reaction of monomers both in solution and bulk. The reformation of the F4D2000–PPO3BMI network structure proceeds with a lower activation energy for the DA reaction, *E_a_,_DA_* = 9 kJ/mol, while the value for rDA reaction *E_a_,_rDA_* = 56 kJ/mol is in accordance with the solution data. The very low activation energy for the DA cross-linking reaction, *E_a_,_DA_* = 7.04 kJ/mol, *E_a_,_rDA_* = 57.9 kJ/mol, was reported also by Polgar et al. [35]. They followed rheologically the reaction kinetics of cross-linking of EPM rubber with DA chemistry after previous decross-linking. The low activation energy was interpreted by diffusion limitation of the reaction in polymer systems. In contrast, Kuang et al. [36] reported a higher *E_a_* of DA type cross-linking of SBR containing furan units due to more difficult diffusion of functional groups in the solid state reaction. We suppose that a low activation energy for the DA reaction is related to the particular topological arrangement of functional groups in the partially broken up structure. The unbroken bonds keep the local structure unchanged and the functionalities from the cleaved bonds in their vicinity are therefore still in a close contact and preserve their location before breaking. As a result, the reconnection of these adjacent functionalities, previously connected, is sterically facilitated and more likely than a random reaction with another functionality.

### 3.2. Network Formation 

Formation of classical polymer networks, arising by stepwise alternating copolymerization reaction, is described by Flory–Stockmayer expression [37] of the classical gelation theory:(*α_A_*)_gel_ * (*α_B_*)_gel_ = 1/(*f_A_* − 1)(*f_B_* − 1)(1)
(*α_A_*)_gel_ = [*r*/(f*_A_* − 1)(*f_B_* − 1)]^1/2^(2)

Gelation of a polymer system occurs at a critical gel conversion *α*_gel_ of monomers A and B. Under ideal random reaction, it depends only on functionality of monomers *f_A_* and *f_B_,* and a stoichiometry of composition *r* (= [B]/[A]). ([A] and [B] are concentrations of the corresponding functional groups.)

The network structure is characterized mainly by cross-linking density *ν*, defined as concentration of elastically active chains, by fraction of the gel *w_g_* and concentration of dangling chains. Usually, the cross-linking density is determined from elastic modulus in rubbery state according to the kinetic theory of rubber elasticity.
*G* = *ν*RTA *w_g_*,(3)
where *G* is equilibrium shear modulus, R is the gas constant, and A is the front factor (A = 1 for affine networks and A = (*f* − 2)/*f* for phantom networks). It is assumed that a transition occurs from the phantom model, applied just beyond the gel-point, to the affine model valid for the fully cured network. 

The theories of network formation describe structure evolution and gelation during a cross-linking polymerization. In addition to classical gelation theory [37], the statistical theories, theory of branching processes [38] or recursive method [39] are used. We applied the theory of branching processes generating branched structures by combination of structural units in different reaction states, the distribution of which is determined from the reaction scheme (see Appendix A). The theory predicts development of cross-linking density/modulus and a fraction of the gel as a function of conversion during formation of reversible networks. The reaction reversibility is taken into account by assuming the equilibrium conversion at any instant of the reaction. 

### 3.3. DA Reversible Networks

The formation of dynamic DA thermoreversible networks is determined by thermodynamic equilibrium of the reversible reaction and its temperature dependence. Contrary to permanent networks, the gelation is governed also by temperature. In addition to the critical conversion at the gel-point, also the gel-point temperature *T_gel_* for the sol–gel transition exists. This critical temperature depends on the gel-point conversion, being thus controlled by functionality of monomers and stoichiometry of composition, and moreover it is governed by temperature dependence of an equilibrium conversion of functional groups. We studied the networks with different functionality of the furan and maleimide monomers, both in the stoichiometric and off-stoichiometric composition, and with a different structure of bismaleimides showing different thermodynamic equilibrium and kinetics of the DA reaction. Moreover, the bismaleimide with a long poly(oxypropylene) chain, PPO30BMI, was also applied to prepare the DA network. This approach makes it possible to tune *T_gel_* in a suitable temperature window and to get a better insight into the formation of reversible DA networks. 

Based on the Flory–Stockmayer expression in Equation (2), the critical conversion of monomers *α_gel_* = 0.58, in the stoichiometric tetrafuran–bismaleimide (F4-M2) networks. Due to the lower functionality of our tetrafuran (*f_w_* = 3.84), the gel point conversion is 0.59. Figure 3 displays the temperature dependence of equilibrium conversions of the maleimide functionality in the three F4-M2 systems differing in the maleimide structure; F4D2000–DPBMI, F4D2000–PPO3BMI, and F4D2000-HBMI. According to the figure, the following theoretical gel-point temperatures of the networks correspond to the theoretical critical conversion; *T_gel_* (F4-DPBMI) = 105 °C, *T_gel_* (F4-PPO3BMI) = 108 °C, and *T_gel_*(F4-HBMI) ~ 120 °C, respectively. Despite the equal functionality of the networks (F4-M2), and thus the equal *α_gel_* in the ideal case, the theoretical *T_gel_* differ by 16 °C. This is a consequence of the maleimide monomer structure governing thermodynamics of the reversible DA reactions and thereby affecting formation of thermoreversible networks. The characterizations of the studied dynamic networks are summarized in Table 1.

The thermoreversibility of networks was determined by rheology measurements and monitoring dynamic storage *G‘* and loss moduli *G“*, as shown in Figure 4. The specimen of a network was heated in the rheometer under plate/plate geometry up to 120 °C in order to decross-link the network. The Figure 4a reveals that *G“ > G‘* at this temperature, implying the liquid character of the sample. The cooling of the melt (blue curves) by the rate 5 °C/min resulted in the network reformation by the cycloaddition DA reaction manifested by increase in modulus. The point of gelation was determined as a crossover of *G’(T)* and *G“(T)* curves (see Section 2.4.2). The more precise gel-point evaluation corresponding to the Winter–Chambon criterion of the critical gel [29] is discussed below. The system F4D2000–DPBMI in Figure 4a gels at *T_gel_* = 86°C and modulus increases with decreasing temperature. The subsequent heating of the network (black curves) leads to a drop of shear modulus *G‘(T)* at ~90 °C. The gel–sol transition, in this case, appears at a higher temperature: 108 °C (see Table 1). The figure shows a very small difference (two dotted lines) in determination of the gel point by using the simplified method from the crossover of moduli compared to the evaluation from the critical value of the loss factor, (tan δ)_gel_ = 1.5, corresponding to the frequency independent loss factor (see Experimental).

The Figure 4c,d reveal that the determined *T_gel_* is dependent on the experimental conditions, i.e. the cooling and heating rates, as it was previously proved [3,13,15]. Gelation of F4D2000–DPBMI is shifted to a higher temperature at a slower cooling rate; *T_gel_* = 92 °C (rate of cooling 3 °C /min), *T_gel_* = 102 °C (1 °C /min), *T_gel_* = 107 °C (0.5 °C /min), which is in a good accordance with the theoretical value (see Table 1). The delay of gelation at a fast cooling rate is a result of the nonequilibrium conditions during DA network formation. The DA reaction is too slow to reach an equilibrium state during a dynamic scan in a rheometer. 

The effect of the cooling/heating rate on the DA reaction progress during the dynamic run is displayed in Figure 5. The network F4D2000–DPBMI was partially broken down at 120 °C and the evolution of conversion during the subsequent cooling and heating processes was determined by FTIR. A short annealing time was applied at each temperature before determination of conversion, simulating thus a dynamic cooling/heating scan of the rate 3 °C /min. The figure illustrates nonequilibrium conditions and large differences of conversions from the equilibrium value (curve 3) determined after the 24 h annealing at the corresponding temperature. The difference is mainly significant during cooling (curve 1), i.e., during network formation. The critical gel-point conversion (*α_gel_* = 0.58) is reached at a lower temperature, leading thus to a decrease in *T_gel_* with increasing cooling rate. Also the reverse gel–sol transition at heating (curve 2) due to the decrosslinking is delayed and the critical temperature for the gel–sol transition in this case is shifted to a higher value with respect to the equilibrium conditions. The difference from the equilibrium, however, is less significant in this case, and therefore also the dependence of *T_gel_* on heating rate is weaker, as it is obvious in Figure 4d. 

The relative rates of the DA reaction and an experimental procedure, i.e. the rate of cooling/heating (*v_exp_)*, play a role as an important kinetic parameter. The equilibrium conditions are achieved only in the case *v_exp_ < v_DA_, v_rDA_*. Otherwise, the dynamic measurement proceeds under the nonequilibrium state. According to the simulation of Scheltjens et al. and Diaz et al [3,15], the equilibrium at the temperature around *T_gel_* in their similar systems could never be met or only at very low cooling/heating rates (<0.5 °C /min). In our experiments, the equilibrium state and equilibrium temperatures *T_gel_* were determined by evaluating both (*T_gel_*)_cool_ and (*T_gel_*)_heat_ received at the cooling and heating scans, respectively (see Table 1). By slowing the experimental rates these values approach from both sides to the equilibrium value. In the case of slow enough experimental scans (*T_gel_*)_cool_ = (*T_gel_*)_heat_ = (*T_gel_*)_equilibrium_. For too slow DA reactions, the *T_gel_* was determined by interpolation (*T_gel_*)_cool_ < *T_gel_* < (*T_gel_*)_heat_. An extreme example of nonequilibrium conditions was observed at formation of F4D2000–PPO30BMI network. Due to a large molecular weight of the bismaleimide PPO30BMI and a correspondingly low concentration of functional groups, the DA reaction is very slow. As a result, no gelation was observed during the dynamic cooling scan of the broken up structure even at a low cooling rate 1°C /min as illustrated in Figure 4b. The experimental determination of gelation is thus affected by the kinetic effect apart from the thermodynamics. 

Gelation of the F4D2000–HBMI system at cooling occurs in Figure 4e already at a temperature *T_gel_* = 108 °C even at the high rate, 3 °C /min. Table 1 shows that the equilibrium state is reached during the dynamic scan at the cooling/heating rate 0.5 °C/min. The determined equilibrium value *T_gel_* = 122 °C well agrees with the theoretical prediction. The network F4D2000–PPO3BMI, split up at 120 °C, undergoes gelation by cooling at *T_gel_* = 85 and 93 °C using the cooling rates 1 °C/min and 0.5 °C/min, respectively (see Figure 4f). Table 1 reveals, however, that the equilibrium was still not set at the rate 0.5 °C/min, because of the slow DA reaction as proved above. Due to a slow network reformation, the gelation was even not observed at the higher cooling rate, 3 °C/min. The equilibrium *T_gel_* = 97 °C was determined by interpolation in the region 93–100 °C (see Table 1). This gel-point temperature is substantially lower than the theoretical prediction in Table 1. In this case, a nonideality of the network formation has to be taken into account. The flexible bismaleimide PPO3BMI is prone to close cycles with the tetrafuran monomer F4D2000, in addition to the intermolecular bonding. The formation of intramolecular bonds at the expense of intermolecular ones does not contribute to a structure growth and a network build-up. It results thereby in delay of gelation and decrease in cross-linking density of a network. Consequently, the total conversion at the gel-point, involving both intermolecular and intramolecular bonds; *(α_total_)_gel_ = (α_inter_)_gel_* + *(α_intra_)_gel_*, is increased in the F4D2000–PPO3BMI system, because only the intermolecular bonds are efficient in the network formation. As a result, *T_gel_* is shifted to a lower temperature with respect to the predicted theoretical value. Gelation at *T_gel_* = 97 °C corresponds to *(α_total_)_gel_* = 0.65 (cf. Figure 3) instead of the ideal value 0.59. One can assume that the extent of cyclization reaches the value *α_intra_*(*= α_total_ –α_inter_)* = 0.06 at the gel-point.

In addition to the tetrafuran–bismaleimide (F4-M2) networks, the hexafunctional furan monomer F6T3000 and the trifunctional maleimide TMIEA were used. The increasing functionality of monomers leads to a decrease in the theoretical *α*_gel_ and increase in the *T_gel_* up to ~145 °C. The Table 1 shows a relatively good agreement of the experimental and theoretical gel-point temperatures in the networks F6-M2 and F4-M3.

Another way of a gelation control is possible by varying the monomers composition. The off-stoichiometric networks (*r* = [M]/[F] ≠ 1) show delay of gelation and form imperfect networks with dangling chains. The off-stoichiometric F4D2000–PPO3BMI network with maleimide in excess, *r* = 1.5, exhibits the decrease in equilibrium *T_gel_* with respect to the stoichiometric network, *T_gel_* = 93 °C (see Table 1).

### 3.4. Isothermal Network Formation

In order to exclude the effect of nonequilibrium conditions, we have followed formation of the DA networks under isothermal conditions. The structure evolution, gelation and postgel growth of the network cross-linking density were determined by monitoring the modulus increase during the isothermal curing. The rate of structure development is governed by the kinetics of the cross-linking DA reaction. The gelation and the postgel evolution as well as final/equilibrium network structure, i.e., the cross-linking density and the gel fraction, however, are affected mainly by the thermodynamics of the reversible reaction. 

We determined formation of the DA networks and evolution of modulus by two procedures. The isothermal time sweep was followed during the reaction of (i) the monomers mixture and of (ii) the partially broken cross-linked structure as described above. 

#### 3.4.1. DA Network Formation from Monomers

Figure 6 shows the growth of dynamic moduli *G’(t)* and *G“(t)* and the simultaneously determined evolution of conversion during formation of the F4D2000–PPO3BMI network at *T* = 70 °C. The gelation, characterized by crossover of *G’* and *G“* curves, occurs in 111 min. The conversion at the gel-point corresponds to *α_gel_* = 0.66 instead of the theoretical value 0.59. This delay of gelation brought about by cyclization is in agreement with that determined from *T_gel_* during dynamic scans. 

The evolution of storage modulus G‘ as a function of conversion is displayed in Figure 7. Both the experimental data and the theoretical curves, calculated by using the theory of branching processes, are shown. The comparison illustrates the effect of cyclization on the network formation. The steep increase in modulus in the curve 1 implies gelation of the ideal random system at *α_gel_* = 0.59. Curves 2 and 3 display the theoretical modulus of the networks involving intramolecular bonding during formation; *α_ι__ntra_* = 0.06 and *α_ι__ntra_* = 0.11, respectively, with the corresponding gel-points at 0.65 and 0.70, respectively. The experimental modulus evolution (curve 4, points) reveals the delay of gelation and lower values with respect to the ideal reaction. The accordance is achieved taking into account the cyclization in the range *α_intra_* = 0.06–0.11. In this case, both the gel-point agrees as well as the postgel growth of modulus. The theoretical curves were calculated using the front factor for the affine network in the Equation (2). The application of the phantom network model in the early postgel stage (see curve 1‘) leads to a negligible difference of the modulus and does not affect the interpretation of the results.

#### 3.4.2. Network Formation from a Partially Broken Structure 

The network formation from a split up structure proceeds after a decross-linking at 120 °C. This procedure simulates a network reformation during self-healing. Figure 8 illustrates the comparison of formation of the network F4D2000–PPO3BMI from a partially broken structure and from the initial mixture of monomers. The gelation and build-up of the network is obviously much faster in the former case. While the mixture of monomers gels at 50 °C in 175 min, the broken structure undergoes gelation within 35 min. In this case, the network reformation starts from the mixture exhibiting already a relatively high conversion as shown in Figure 2.

The effect of temperature and the bismaleimide structure on gelation and a structure evolution is illustrated in Figure 9. The fast network build-up from a broken structure is an important parameter for a network self-healing. Figure 9a displays the structure growth of the F4D2000–PPO3BMI network at different temperatures. Both kinetics and thermodynamics of the DA reaction play a role determining the optimum temperature for the network development. The rate of network formation is accelerated at increasing temperature up to 80°C. The gelation, manifested by crossover of *G’* and *G”*, sets in 7 min and the modulus increase is the fastest at this temperature. At a higher temperature, the rDA breaking reaction becomes operative, resulting thus in delay of gelation and slowing down the structure development. Decrease in equilibrium conversion at a higher temperature leads to a lower final cross-linking density of the network. At *T* = 100 °C the system does not gel because *α*_e_ < *(α_inter_)_gel_*. The build-up of the network F4D2000–PPO30BMI in Figure 9b is much slower due to the slow DA kinetics. The optimum temperature in this case is *T* = 70 °C; the system gels in 100 min. In contrast, an almost instantaneous gelation happens in the case of F4D2000–HBMI after cooling (see Figure 9c).

The Figure 9c reveals that the rate of network reformation, characterized by the inverse gelation time and rate of the modulus growth, is controlled by structure of bismaleimides and decreases in the series HBMI > DPBMI > PPO3BMI > PPO30BMI. This rate depends on the DA reaction kinetics (the DA rate decreases in the same order, see above), and the extent of the structure splitting at 120 °C (*α_120_*), i.e., the distance from the critical conversion, *α_Δ_ (= α_gel – _α_120_*). The distance *α_Δ_* ~ 0 for HBMI containing system, because the stable HBMI network shows a very incomplete decross-linking at 120 °C, just to undergo the gel–sol transition; the conversion reaches the value around the critical one (cf. Figure 2 and Figure 3). The „DPBMI“ and „PPO3BMI“ networks are more broken up because of a lower equilibrium conversion, *α_Δ_* ~0.13 and ~0.20, respectively, since the latter network exhibits an increase in *α_gel_* due to cyclization. The gel times determined from Figure 9 correspond to the following critical conversions evaluated from Figure 2; *α_gel_* ~ 0.59 for „DPBMI“ network, in agreement with the theory, and *α_gel_* ~ 0.65 for „PPO3BMI“ network. Slow gelation of PPO30BMI containing system is a consequence of the small DA reaction rate. 

### 3.5. Modulus and Sol Fraction of DA Reversible Networks

In addition to the rate of gelation, the cross-linking density and the corresponding modulus, as well as the gel fraction are the main parameters characterizing a network formation. Due to the temperature-dependent conversion in thermoreversible networks, the cross-linking density and the gel fraction are functions of temperature and the modified expression holds for the equilibrium modulus: *G_e_* = *ν*(*T*)ART*w_g_*(*T*)(4)

The G‘/T plots of the homogeneous networks F4D2000–PPO3BMI, F4D2000–PPO30BMI and F4D2000-HBMI are illustrated in Figure 10a. Moreover, the corresponding theoretical curves are included as dash lines. For the theoretical calculation, the equilibrium conversion is taken at any temperature. At increasing temperature, the theoretical equilibrium moduli decrease due to the diminishing conversion of the DA reaction. In contrast, the experimental moduli do not decline or only slightly until 70 °C. The reversible networks are in equilibrium state at ambient temperature, however, at higher temperatures the conditions are nonequilibrium during the temperature sweep. The DA/rDA reactions are slow in comparison to the rate of heating (2 °C/min) and the conversion does not reach the equilibrium value. Two experimental equilibrium points at higher temperatures, received by isothermal annealing of F4D2000–PPO30BMI, are included in the figure for a comparison with the theory.

The experimental moduli are in agreement with the theoretical ones for PPO30 containing network and in a relatively good accordance for the „HBMI“ network. The lower value of the experimental modulus in the „PPO3“ network is brought about by cyclization as discussed above. The agreement of the equilibrium modulus with the theory at room temperature is achieved assuming the intramolecular reaction in the extent *α_intra_* = 0.04 (curve 1‘c). 

Consequently, four independent results provide an evidence of cyclization in the PPO3BMI containing network. The fraction of intramolecular bonds was determined to be in the range 0.04-0.12 from the shift of *T_gel_*, from the critical *α_gel_* value at isothermal network formation, from the postgel growth of modulus, and from the equilibrium modulus of the cured network at room temperature. No cyclization occurs in the „PPO30BMI“ network because the corresponding ring in this case is too large. The relative tendency to cyclization wih respect to the intermolecular reaction depends on the chain flexibility and the size of the smallest possible cycle. With increasing size of this ring the cyclization probability significantly decreases [40]. Neither HBMI network likely undergoes cyclization. There is a phase separation of the aliphatic bismaleimide and the propylenoxide based furan monomer at the early stages of the reaction (see Appendix A), and therefore the intermolecular reaction is preferred with respect to the intramolecular one. The cyclization with the rigid DPBMI molecule is also unlikely. 

The reversible networks show a high sol fraction, which limits their materials applicability. The equilibrium values *w_S_* (*=1 − w_g_*) cannot be determined experimentally, because an extraction of the monomers and sol fraction from the network leads to a continuous shift of the reversible DA reaction towards monomers. The F4D2000–PPO3BMI network completely dissolves in DMSO at 50 °C, despite being far below *T_gel_*. The problem of the sol fraction in reversible networks is not investigated in literature, despite being extremely important for a materials application. Taking into account the reasonable agreement of the theory with experimental moduli, one also assumes that w_S_ values could be well predicted. The theory was used to provide the information on this important aspect. The theoretical w_S_ as a function of temperature are shown in Figure 10b. In the case of the „HBMI“ network, the sol fraction is negligible up to 80 °C. However, the PPO3BMI containing network involves at 80 °C already a quite high fraction of the sol (>0.20) due to the lower equilibrium conversion and the cyclization. The effect of the DA reaction thermodynamics is revealed by comparison of the curves 1 and 2 for PPO3BMI and HBMI containing networks, and the influence of cyclization is displayed by the curve 1‘, characterizing „PPO3BMI“ network undergoing 6 % of intramolecular reactions. 

### 3.6. Dynamics of the Transient Networks

The thermoreversible networks exhibit a transient character as proved by the rheology frequency sweeps of the networks F4D2000–DPBMI and F4D2000–HBMI shown in Figure 11.

Three regions are observed in the curves of the storage modulus as a function of frequency in the F4D2000–DPBMI network (Figure 11a); a wide region of the rubbery plateau, a flow region at a low frequency, and vitrification at a high frequency. This network exhibits a relatively high glass transition temperature, *T_g_* = 40 °C, and therefore at 50 °C and high frequencies, the modulus is affected by the glass transition.

The networks show a rubbery modulus plateau with the storage modulus *G‘* independent of frequency at temperatures *T < T_gel_*, where *G‘ > G“*. The plateau modulus decreases with increasing temperature due to diminution of the conversion. In the network containing DPBMI, the modulus plateau is observed at *T* = 95 °C, however, at 100 °C, the plateau disappears and the moduli *G’(ω)* and *G“( ω)* show the similar scaling at higher frequencies, *G‘(ω)~G“(ω)~ω^n^* (n = 0.64). This state corresponds to the Winter–Chambon criterion for the gel-point [29]. The critical gel occurs in the temperature region 100–105 °C, which is in a good agreement with the *T_gel_* equilibrium value determined by the simplified method from the crossover of *G’*and *G“* measured at a constant frequency (see Figure 4, Table 1). At a higher temperature, *T* = 110 °C, the polymer exhibits liquid-like behavior; *G“ > G‘*, *G‘~ω^2^, G“~ ω^1^.* While the networks with DPBMI a PPO3BMI maleimides show a similar dynamic behavior, the „HBMI“ network exhibits a higher thermal stability. The broad plateau, characterizing the solid network state, exists even at 110 °C and only at *T* = 120 °C the polymer is approaching the critical state and flows.

The liquid-like behavior appears, at the low-frequency, even at temperatures well below *T_gel_*, i.e. far above the gel-point conversion. The terminal relaxation of the network is a macroscopic demonstration of a network decross-linking by the rDA reaction. The relaxation time of the network *τ_r_* was determined from the frequency crossover *ω_C_ (τ_r_ = 2π/ω_C_*_)_ in Figure 11. The crossover of *G’*(*ω*) and *G”*(*ω*) is gradually shifted to a higher frequency at increasing temperature, i.e., the relaxation time *τ_r_* becomes shorter, revealing thus the reducing stability of the network. The network connectivity is thus broken at shorter times when approaching to *T_gel_*. Also this characterization of the network dynamics reveals the highest stability of the HBMI containing networks. The relaxation times are similar in the case of „DPBMI“ and „PPO3BMI“ networks, but they are much longer for the „HBMI network“. At 90 °C the following values were determined; *τ_r_* (DPBMI) = 400 s and *τ_r_* (HBMI) > 2000s. Even at 100 °C and 110 °C the HBMI network is relatively stable with *τ_r_* = 1000 s and 170s, respectively.

The dynamics of the DA networks thus involves the thermodynamically governed network breaking at higher temperatures due to the temperature dependence of the equilibrium conversion. Moreover, there is the kinetically controlled time dependent network disconnection at any temperature related to lifetime of the reversible bond. At a high temperature, the dynamic character becomes more pronounced because of higher rates of the bond forming/breaking reactions and a shorter bond lifetime. The relative rates of the DA reaction and experimental dynamic measurements govern the time scale of the transient region from an elastic network behavior to the liquid state. At low temperatures (50 °C in DPBMI and 90 °C in HBMI), the reaction rates are relatively too small, the networks are more stable and the transition is shifted to long time scales to be observed during the experimental measurement conditions. 

### 3.7. Design of a Self-healing Network 

The understanding of the reversible networks formation is a basic step to optimize the self-healing procedure and to design a proper structure of a self-healing network. For an efficient and promising self-healing, the polymer network should be healed in a suitable temperature window and the healing, including structure reformation, should be rapid. Moreover, the incomplete conversion in the reversible networks results in a reduction of modulus and a high fraction of the sol even at ambient temperature. The corresponding poor mechanical properties and leaching of monomers present the significant drawback of the reversible networks. Therefore, the high stability of the network at ambient temperature is desirable.

The DA network breaking/formation is affected by morphology of a system and its physical state. In order to design the system for self-healing at an appropriate region of temperatures (20–130 °C), the glass transition temperature *T_g_* and gel-point temperature *T_gel_* of the network must be tuned by controlling the monomers structure and functionality. Low enough *T_gel_* (<130 °C) prevents participation of irreversible side reactions and low *T_g_* (<50 °C) excludes vitrification and a significant deceleration of the DA reaction in a network. The optimum system exhibits a rubbery modulus plateau in the proper temperature range providing enough space for a structure reconstruction at cooling. The applicability of the studied networks is obvious from DMTA plots in Figure 12, characterizing roughly their morphology. The F4D2000–PPO3BMI and F4D2000–PPO30BMI networks are homogeneous and rubbery at ambient temperature. Their structure evolution could be well described by theory and thereby they serve as ideal systems to get an insight into the DA network formation. Also HBMI containing network and the heterogeneous network involving DPBMI show a rubbery plateau despite a relatively high *T_g_* in the „DPBMI“ network shortening the applicable rubbery plateau region. In contrast, the networks from monomers of a higher functionality: F6T3000-DPBMI (F6-M2) and F4D2000-TMIEA (F4-M3) exhibit too high *T_gel_* (>130 °C). The network F3FGEFA-PPO3BMI, containing the short trifuran monomer instead of the long flexible F4D2000, is also unsuitable due to the high *T_g_* preventing thus a DA network reformation because of vitrification. 

The network F4D2000-HBMI was shown to be the best system from the point of the self-healing view. The alkyl substituent in the maleimide increases stability of the DA adduct, while it does not prevent reversibility of the reaction. This dynamic network is very stable at ambient temperature showing the conversion *α_e_* > 0.95, and therefore it exhibits a relatively high modulus and a negligible sol fraction, *w_S_* < 0.01(see Figure 10b). It also undergoes a very rapid structure healing as determined by simulating the healing procedure and following network reformation from a broken structure (see Figure 9c).

## 4. Conclusions

The formation of the reversible furan–maleimide networks was controlled by various parameters. The critical gel-point temperature, *T_gel_*, as well as the network properties were tuned in a wide range by the maleimide structure, by functionality of monomers and their molar composition in the network. The substitution of maleimides with alkyl (HBMI), aromatic (DPBMI) and polyether (PPOBMI) substituents affects differently the kinetics and thermodynamics of the thermoreversible DA reaction, and thereby the formation of dynamic networks. Both dynamic nonequilibrium and isothermal equilibrium procedures of a network formation revealed the significant effect of the maleimide structure on gelation of the reversible systems. The *T_gel_* in the range 97 to 122 °C were determined for the F4-M2 type networks with different structure of maleimide monomers. The network F4D2000–HBMI, involving the alkyl substituted maleimide HBMI, shows the highest equilibrium conversion at the reversible reaction resulting in the highest thermal stability of the network, while still in the applicable temperature window, the highest modulus and the smallest fraction of the sol. At the same time, however, the reaction reversibility is not limited.

The *T_gel_* value is the crucial parameter characterizing formation of thermoreversible networks. However, the experimentally determined *T_gel_* from the dynamic *G‘/T* run suffers from nonequilibrium conditions and depends on the DA kinetics and its relative rate with respect to the experimental measurement. In order to determine the equilibrium *T_gel_*, the approach eliminating the kinetic effect was applied. The determined values of *T_gel_* are in accordance with the temperature region corresponding to the critical gel state according to the Winter–Chambon criterion based on the isothermal frequency sweeps of moduli. The networks exhibit transient character and the terminal relaxation at low frequencies even at temperatures *T < T_gel_*, i.e. at *α > α_gel_*.

Theory of branching processes was used to predict the structure development during polymerization of homogeneous systems. In the networks F4D2000–PPO30BMI and F4D2000–HBMI, the gelation, postgel structure evolution, followed as development of modulus, and the final/equilibrium modulus agree with the theory. The comparison of the experimental data with the theory provided an insight into the mechanism of formation of F4D2000–PPO3BMI network and revealed the cyclization reactions. Moreover, the theory provides an important information on the experimentally unmeasurable sol fraction in the reversible networks.

The acquired results and study of the reversible network formation from a partially split up structure were used to design a proper structure of a self-healing system. The F4D2000-HBMI network was found to be the most promising. It exhibits the suitable temperature window for healing (20–130 °C) manifested by a wide rubbery modulus plateau. Due to the high equilibrium conversion and fast DA kinetics it shows the extremely rapid structure reformation/healing, as well as the relatively high modulus and the negligible sol fraction, which are important properties of the high performance self-healing polymers. The high performance self-healing reversible network still is a challenge. Reinforcement with nanofillers and a study of reversible nanocomposite networks is underway. Moreover, for a self-healing material, the shape-persistent structural motif is incorporated by preparation of dual networks consisting of reversible–irreversible structures.

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
