# Peer review of "Control of Gelation and Properties of Reversible Diels–Alder Networks: Design of a Self-Healing Network"

_polymers, 2019, doi:10.3390/polym11060930_

Round 1

Reviewer 1 Report

The paper discusses the reversible network formation using combinations of furan-functionalized Jeffamines and different maleimide crosslinkers. The manuscript describes interesting insights in controlling the reversible network formation via the Diels-Alder cycloaddition reaction, however the scientific level of the research (kinetics and fundamental polymer science) is lower than current literature.

Improvements of the scientific discussion may lead to an interesting added value for scientific literature in this interesting field.

The reaction time for the furan functionalization of the Jeffamines using FGE is very short. Epoxy-amine reactions typically require much longer times at more elevated temperatures. The 1HNMR spectrum clearly shows remaining unreacted epoxy groups (2-3 ppm). What is the extent of reaction for the different furan-functionalized compounds?

The main text refers to 1HNMR spectra of the FMA, HBMI and other maleimides in the supplementary information, while those are not included. These measurements should be included and extents of reactions need to be calculated.

Section 2.4.2 is called "Dynamical Mechanical Analysis" while it should be "Dynamic rheometry" The size of the plates and strains or deformations used for the oscillatory measurements should be detailed.

The DMA technique and measuring conditions should be discussed in the experimental section.

Line 222: The activation energies are indeed very low compared to literature. References 11 and 12 provide kinetic parameters for similar polymeric systems (DPBM-based) that are very different. Moreover, more recent literature (https://pubs.rsc.org/en/content/articlelanding/2019/py/c8py01216d#!divAbstract) elaborately describes the effect of stereochemistry on the reversibility of the Diels-Alder reaction for similar polymeric systems (aliphatic maleimides). It addresses the shortcomings of FTIR to distinguish between endo and exo adduct formation. The current manuscript should either improve the credibility of the kinetic results or correctly address the limitations of the obtained kinetic paramets with respect to literature: stereochemistry not taken into account & comparison to exisiting literature. I want to stress that accurate kinetic modelling is require for credible kinetic simulations for the further studies in the manuscript.

Line 289: according to ref 27 the reactivity of HBMI and PPO3BMI should be similar, which does not seem to be the case in the current manuscript. What would be the reason?

Line 304: in the elastomeric systems with low crosslink densities there should not be any topological effects and definitely no diffusion limitations for the Diels-Alder reaction, especially at the studied temperature (room temperature and above). The following discussion does not provide sufficient justification for the low activation energies.

Table 1: why is there a "92" below the first "equilibrium"? The theoretical gel point of the off-stoichiometric mixture is missing in the table.

In the experimental section the gel point is defined as the frequency independence of the complex modulus (Winter Chambon), however in the results the gel point is determined as the cross-over between G' and G" . This is inconsistent and should be corrected. The former definition is the most correct one and should thus also be used in the results to determine the gel points (i.e. frequency independency of loss angle).

Line 435: A literature study on a similar system (http://specificpolymers.fr/medias/publications/2017-06.pdf), using the same bismaleimide and Jeffamine D400-based furan, shows that the gelation does occur at the theoretical value. Either the theoretical value has not been determined correctly, or something else is going wrong with the gel transition. Please check the polymer formulation etc. There is no reason for such a conservative system not to follow gelation theory.

Figure 6 adjust formatting of the axes.

Paragraph from line 713 until 739 does not concern new results, but describes information previously described in literature and should be shortened and moved to the introduction with adequate references (e.g. https://pubs.rsc.org/en/content/articlelanding/2013/cs/c3cs60046g#!divAbstract and/or https://www.sciencedirect.com/science/article/pii/S0032386117305359)

Author Response

The reply is uploaded.

Reviewer 2 Report

In this manuscript, Matejka et al. thoroughly investigated the structure-property relationships of Diels-Alder type polymer networks by designing and tuning the monomer structures (furan diene and maleimide dienophile). According to their data, various spacers of maleimides including alkyl, aromatic, and polyether had dramatic effects on the kinetics and thermodynamics of DA reaction. Moreover, they evaluated the gel points of each network as well as observing that F4D2000-HBMI gel possessed the highest thermal stability among all the networks. In general, the design of this study is logical. The data are sufficient to support their conclusions. The writing is clear to be followed. Given the depth and importance of this study, I would like to recommend acceptance of this article in Polymers. However, the authors should address my comments to improve the manuscript before publication can be granted.

Comments

1. Page 3, lines 94-97, please add the units for the molecular weights.

2. The authors need to provide the full characterizations of monomers as shown in Scheme 1. For example, the NMR and FTIR of F3FAFGE and F6T300 should be shown in supporting documents.

3. The formatting and the resolution of chemdraw should be uniform. Some are big and some are very small in the same figure. A better drawing would help readers to catch the structures more easily.

4. In scheme 1, the chemdraw of PPO3BMI is incorrect. “N” is missing in the maleimide rings at both two ends of the molecule. Please fix them.

5. The references for DA reactions in introduction are a bit old. Some recent and important references regarding DA chemistry should be added in addition to ref. 6 and 7. Highly suggest adding (1)Sumerlin at al. Progress in Polymer Science, 2019, 89, 61-75. ; (2) Nature Chemistry, 2017, 9, 817-823. and (3) Chemical Science, 2014, 5, 4646-4655.

Author Response

Answers are uploaded.

Round 2
